# Assessment of Water Availability and Environmental Influence on People's Lives in a Small Basin in the Hinterland of Pernambuco, Using the SUPer and UAV

Gabrielly Gregório da Luz [1,*,†], Rodrigo de Queiroga Miranda [2,3] and Josicleda Domiciano Galvíncio [3,*]

1 Department of Geography, Federal University of Pernambuco, Recife 50670-901, Brazil
2 Department of Animal Science, Universidade de Manitoba, Winnipeg, MB R3T 2N2, Canada; rodrigo.qmiranda@gmail.com
3 Department of Geographical Sciences, Federal University of Pernambuco, Avenue Professor Moraes Rego, 1235 Cidade Universitária, Recife 50670-901, Brazil
* Correspondence: gabrielly.gregorio@ufpe.br (G.G.d.L.); josicleda.galvincio@ufpe.br (J.D.G.)
† Bachelor's degree in Geography from the Federal University of Pernambuco, currently pursuing a Master's degree in the Development and Environment Postgraduate Program at the Federal University of Pernambuco.

**Abstract:** Water scarcity is a worldwide concern considering that water is a limited resource and essential for life. In Brazil, approximately 30% of its population lives in a semi-arid region covering about 20% of the country's territorial extension, which is one of the areas that most suffers from a lack of water. The lack of water, mainly in the northeast of the country, has been a problem for years, as people who live in this territory suffer for months from the poor distribution of this resource, which increases the degree of inequality between the regions of the country. The research aims to show the effect of the hydrological cycle on the quality of vegetation and how such processing can end up affecting people's lives and the environment. This study carried out a temporal analysis from 1961 to 2021. The hydrological model system used to assess water availability was the Pernambuco Hydrological Response Units SUPer-System. UAV (Unmanned Aerial Vehicles) was used to view the relationship between living and environmental conditions. The results showed a difference between the water balance today and in the future due to climate change. Thus, it is concluded that climate change will have different impacts at a small scale as well as on people's living conditions as a result of different characteristics of the environment. It is very important to carry out studies on a detailed scale to provide better public policies for mitigating the effects of climate change on people's lives.

**Keywords:** annual water dynamics; hydrology modeling; climate change

## 1. Introduction

The changes arising from global environmental change have generated significant interest within both governmental [1] bodies and the scientific community, as it has become increasingly evident that this phenomenon poses a threat to numerous populations across the globe [2,3]. The utilization of water is contingent upon its availability, and in semi-arid regions, this factor demonstrates temporal and spatial variability, particularly within small watersheds [4]. According to the National Water Agency (ANA) [5], the PISF (São Francisco River Integration Project) aims to enhance the security of water supply in reservoirs within the semi-arid region. Consequently, monitoring the water availability of semi-arid river basins is of utmost importance to ensure water security for the population [6].

Water security is intricately linked to chronic actions and extreme events [7]. In semi-arid areas, water security is perpetually a contentious and vital subject. A comprehensive understanding of the region's hydrological characteristics is imperative for informed public policy decisions. On certain occasions, acquiring detailed on-site data can be financially prohibitive. Therefore, hydrological modeling becomes a valuable and essential tool for

accurately assessing water conditions in a given region [8]. Thus, hydrological modeling plays a pivotal role in the management and decision-making processes related to water security in semi-arid areas, aiding in the optimization of the utilization of the limited water resources available.

The hydrological model Soil Water Assessment Tools-SWAT is very widespread in Brazil and due to the need for adjustments in the results obtained through modeling and the observed data, it is necessary to carry out a calibration. Different studies have been developed to calibrate the swat model to Brazil [9], including [10–12], and for the northeast of Brazil [13], and for the Pernambuco basins [14–16]. The intention was and is to provide a system that can facilitate the monitoring and planning of watersheds. This system is called the SUPer-System of Hydrological Response Units for Pernambuco.

Watersheds can be regarded as natural systems that play a pivotal role in the hydrological cycle. It is essential to underscore that water flow may constitute the most critical component of the hydrological cycle and can be influenced by climate change and human activities [17]. Climate change and human activities are determinants of both the quantity and quality of water, and the analysis of these two factors is exceedingly important in the formulation of public policies [18]. In Brazil, the River Basin Committees (CBHs) were established to implement the provisions of the Water Law [19]. These committees consist of individuals with diverse interests who advocate for, analyze and deliberate on the programs included in the Basin Plans, representing a significant initiative for the entire civil community. A hydrographic basin is responsible for receiving rainwater that falls on the surface, which can be captured and stored in reservoirs, lakes, soil and underground aquifers, as demonstrated by [20].

Studies that investigate the behavior of precipitation and flow in a watershed have garnered the attention of researchers worldwide [21]. The quantity and temporal distribution of precipitation are of utmost importance in determining the flow regime of rivers, which directly impacts water availability. Studies such as [22] elucidate the significance of precipitation in conjunction with extreme events that have a direct bearing on the lives of people residing in a specific watershed. Another highly pertinent process within the hydrological cycle is evapotranspiration, which has become the second most critical variable within the hydrological cycle [18]. When the atmospheric demand for evapotranspiration (ET) surpasses the supply of precipitation, it leads to indicators of water deficit, potentially exacerbating social vulnerability through reduced water availability and limited access [23]. Understanding the quantification of ET is imperative to grasp the dynamics of the interactions between atmospheric energy and the Earth's surface, as well as regional water balances in watersheds [24].

The São Francisco River Integration Project (PISF) was executed by the Ministry of National Integration [25]. It stands as one of the most significant undertakings in the semi-arid region aimed at ensuring water security, with the entire scientific community engaging in discussions regarding the project's efficacy [26]. Concerning the physical aspect of the PISF, it encompasses the construction of two axes that integrate the São Francisco River with intermittent hydrographic basins: the NORTH and EAST axes [27]. The arrival of water from the Velho Chico, via the east axis, raised expectations of potential increased access to water for the municipalities within the channel [26]. This research aspires to elucidate the profound impact of the hydrological cycle on the lives of individuals and the environment.

## 2. Materials and Methods

### 2.1. Area Study

The hydrographic basin under study has an area of 236.02 km$^2$. It is a sub-basin of the Pajeu hydrographic basin. This area is in the hinterland of Pernambuco with semi-arid weather conditions, Figure 1. It is located next to the São Francisco River and is under the impact of the transposition of the São Francisco River. Part of its basin covers indigenous areas, as seen in Figure 1B. The basin is located at UTM coordinates −8.526 and −38.124,

the water balance rations in the basin study is streamflow/precip, which is 0.244; the baseflow/total flow is 0.545; the surface runoff/total flow is 0.455; the perc/precip is 0.163; and the ET/precipitation is 0.695. The land use in the basin is SPAS-summer pasture and CRWO-cropland/woodland mosaic. The geomorphology is shown in Figure 1C.

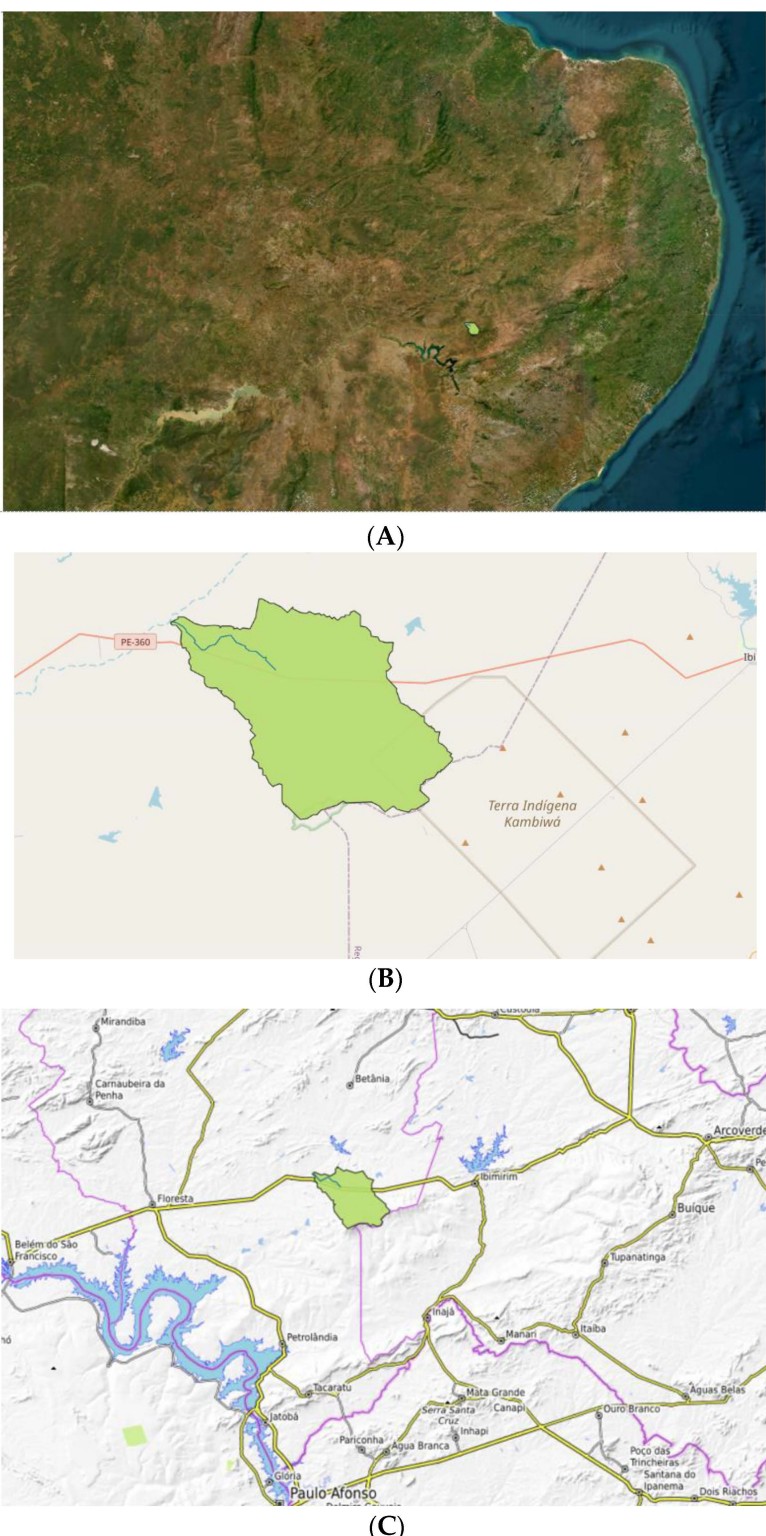

**Figure 1.** Location of the hydrographic basin 73, in Pajeu. (**A**) Spatial localization, sub-basin 73. (**B**) Detail scale, sub-basin 73. (**C**) Geomorphology.

In this study, 61 years of daily-scale data were utilized, covering the period from 2016 to 2021. These daily data are available within the model. Daily hydrological cycle data contribute to the effective management of water resources within a region. This encompasses the monitoring of water levels in rivers, lakes and aquifers, as well as the measurement of precipitation and evaporation rates [28]. Such information is indispensable for the planning and allocation of water resources for potable water supply and agricultural irrigation. To warm up the SWAT-Soil Water Assessment Tools model, five years were used. The SWAT model that was used for water balance modeling was the 2012 version [12].

The SWAT model (Soil and Water Assessment Tool, available free of charge at http://swat.tamu.edu accessed on 12 May 2023 has been gradually expanding its borders, as it has a profile of varied models for the interests of both private companies and public bodies. SWAT is a semi-distributed model in which a basin target river basin is divided into several sub-basins, which in turn are subdivided into hydrological response units (HRUs) that consist of areas homogeneous in relation to land use and cover, relief and soil characteristics. The HRUs are represented in the model as a percentage of the sub-basin.

SWAT requires observed data on four essential components: relief, climate, soils, and land use and cover. Relief: spatial relief data were obtained using the data from EMBRAPA Relief (https://www.cnpm.embrapa.br/projetos/relevobr/ accessed on 12 May 2023), which provide corrected products from the NASA SRTM (Shuttle radar topography) mission, which aims to map the topography of the Earth's surface. The data from this model are in image format, whose pixels have a spatial resolution of 90 m and altitude (m). Climate: the climate data of the series, i.e., precipitation, global radiation, humidity relative air temperature or dew point temperature, average air temperature, temperatures air maxima and minima, and wind speed, were obtained for the years spanning from 1961 to 2021 through the INMET database (National Institute of Meteorology; http://www.inmet.gov.br/projetos/rede/pesquisa/inicio.php accessed on 12 May 2023). Then, only the station of Floresta located in the state of Pernambuco was selected. We used average monthly values simulated and observed in Floresta with a longitude of $-38.64$ and latitude of $-8.61$. The station code is 48860000 to the runoff obtained by the Brazilian water agency. The Floresta municipality was chosen because it was the closest station to sub-basin 73 under study, and so the model would be better calibrated and adjusted for this area. Because of this, here, we validated the SUPer data with observed data. To evaluate the calibration of the model, the determination coefficient was used.

Soils: data spatial characteristics of soil characteristics were obtained through two databases: (i) the spatial mapping of soils came from the Agroecological Zoning of the State of Pernambuco (ZAPE). These data are available in the form of a map with a scale of 1:250,000 on the website http://www.uep.cnps.embrapa.br/zape accessed on 12 May 2023, and (ii) the data relating to the physical-chemical measurements of each type of soil were from the EMBRAPA Soils database (https://www.bdsolos.cnptia.embrapa.br/consulta_publica.html accessed on 12 May 2023). Land use and cover: initial usage and coverage data were obtained through Landsat TM 5, ETM+/Landsat 7 and OLI/Landsat 8 with the 1975–2013 series available at https://earthexplorer.usgs.gov/ accessed on 12 May 2023 and www.dgi.inpe.br/CDSR/ accessed on 12 May 2023. The products were downloaded based on radiometric quality and the absence of cloud cover (<10%), and created eight annual land cover maps (1975, 1993, 2001, 2003, 2005, 2007, 2009 and 2013) using 42 images and the available GDAL (Geospatial Data Abstraction Library) at http://www.gdal.org/ accessed on 12 May 2023.

At this stage, all data were formatted for input into the SWAT model, and the project was created with the help of the ArcSWAT tool available for ArcGIS software in http://swat.tamu.edu/software/arcswat/ accessed on 12 May 2023. The usage maps were processed using the tool SWAT2009_LUC_64bit.exe, which is a preprocessor that implements the land cover dynamics in SWAT. Details about the software can be obtained from. Each project corresponds to a single basin. ArcSWAT has been configured to delimit sub-basins or drainage areas of approx. 10,000 hectares, generating sub-basins and HRUs (Hydrological

Response Units); Range Brush was the vegetation type available in the SWAT database and chosen to represent the Caatinga, due to its proximity in terms of physical characteristics. Initially, the data available in the database regarding the Range Brush were not modified, except for the vegetation class that was modified for the warm season annual legume to avoid the dormancy of the vegetable vegetation. It was used for five years for the warm-up of the model. For this year's warming, SWAT simulated climate data using the weather tool generator.

### 2.2. Model Calibration and Validation

Software: The model was calibrated using SWAT-CUP software 2012 (SWAT Calibration and Uncertainty Programs). SWAT-CUP is software that integrates the outputs of the SWAT model and five calibration algorithms, including SUFI2 (Sequential Uncertainty Fitting 2; Abbaspour; Johnson; Van Genuchten, 2004), which stands out for its speed and accuracy in processing and consists of three major steps: modifying the values of SWAT inputs, running SWAT and extracting the desired output values.

### 2.3. Statistical Analyses

The model performance was obtained from observed and simulated values using the following statistical methods: Nash-Sutcliffe (NS) efficiency coefficient. The NS varies from ∞ to 1 NASH; SUTCLIFFE, 1970 where 1 indicates a perfect simulation.

After calibration, the model was uploaded to the SUPer-System of Hydrological Response Units for Pernambuco, link: https://super.hawqs.tamu.edu/ accessed on 12 May 2023.

In this study, the SUPer system (System of Hydrological Response Units for Pernambuco) was used as a simulation model for the quantification and quality of water in the basin.

The methodological basis of SUPer is the SWAT. SUPer has been used to simulate numerous basins in the state of Pernambuco for soil and water resource assessments. SUPer is an efficient model that continuously simulates a wide variety of watershed processes for a defined recording period. SUPer allows you to estimate soil moisture in spatial and temporal detail. Thus, it is believed that the SUPer is very useful in indicating disasters that have a good relationship with water balance and soil moisture.

In the SUPer, the model was run in daily scale, the set-up/warm-up was five years and the SWAT model version was SWAT2012. The project in SUPer to sub-basin 73 with the HRU-Hydrology Response Unit has seven.

### 2.4. Scenery

1. Actual
2. Semi-arid decrease in rainfall by 15% and increase in temperature by 2 Celsius (climate change)

The criteria for the scenario setup were without any changes to consider the current situation. The second criterion considers the study by [29].

The hydrological model system that was used to assess water availability was the SUPer-System of Hydrological Response Units for Pernambuco. In addition, this system helped to understand the changes that occurred in the sub-basin studied. The study can obtain accurate and necessary information to understand the dynamics of the hydrological.

For a better analysis of the impacts of climate change on the environmental conditions of the vegetation, the analysis of biomass by land use was used, where SPAS-summer pasture and CRWO-cropland/woodland mosaic, soil types are RQ—Risk Quotient (represents relative activities of aerobic and anaerobic microbial metabolism) and TC—eutrophic soil and slope is 0–3, 3–8, 8–20 and 20–45.

In this study, aerial photography images obtained with UAV (Unmanned Aerial Vehicles) in 28 September 2023 were used to visualize changes in vegetation and surface temperature in areas with changing conditions in the São Francisco River water diversion project. That is, an area with water entering the hydrological system.

The findings served to investigate the environmental change in the sub-basin and how this phenomenon is interconnected with the resident population in the study area.

## 3. Results

### 3.1. Calibration of the Basin

Before uploading to the SUPer system, the model was calibrated using the SWAT model. It is noted that the calibration control points that the model presented were $r^2$ above 0.67, as shown in Figure 2. Different studies have already been developed for the basin with the aim of contributing to the better management of water resources in the basin [30].

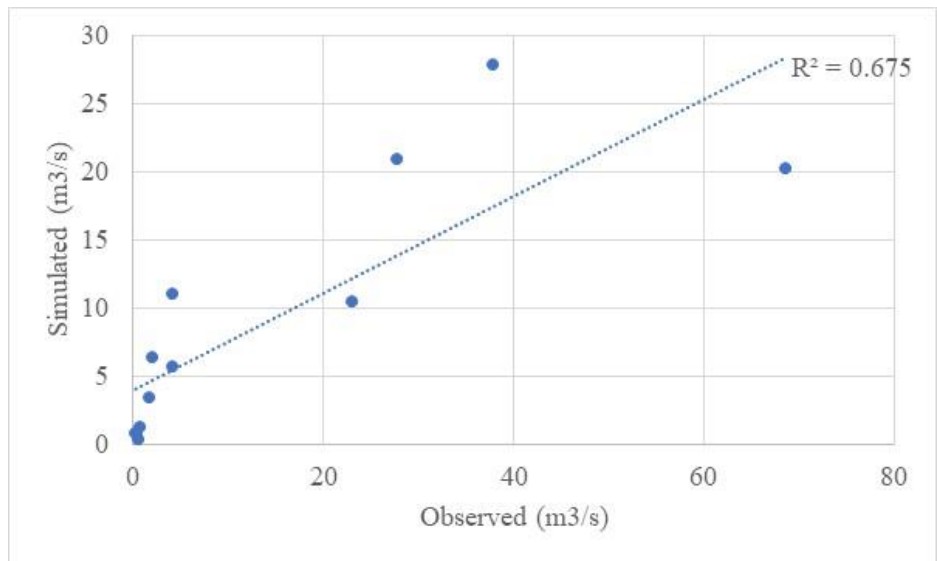

**Figure 2.** Relationship between runoff observed and simulated for the Pajeu basin.

The average annual water balance of basin 73 presents an average annual precipitation of 661 mm, surface runoff of 73 mm, lateral flow of 20 mm, return flow of 67 mm, evaporation and transpiration of 459 mm and potential evapotranspiration of 1788 mm, as seen in Figure 3. The statistics of the average annual water balance is shown in Table 1. Note that 69% of precipitation is due to actual evapotranspiration. The baseflow is 55% of the total flow. The baseflow is the lateral flow and return flow. The total flow is 161.34 mm. In the first layers of the soil, infiltration/plant uptake/soil moisture redistribuition occurs, which transforms in the lateral flow. The streamflow is the total flow. The streamflow was 24% of the precipitation. That is, 24% of the precipitation is transformed into runoff. The percolation to shallow aquifer is 16% of the precipitation. The baseflow is 13% of the precipitation. The surface runoff is 11% of the precipitation. When we simulate climate change, the water balance is greatly altered. There is a significant decrease in the baseflow, as shown in Figure 4. I draw attention to the baseflow because it is very important in semi-arid areas.

**Table 1.** Water balance ratios.

| | |
|---|---|
| Streamflow/Precip | 0.244 |
| Baseflow/total flow | 0.545 |
| Surface runoff/total flow | 0.455 |
| Perc/precip | 0.163 |
| Deep recharge/precip | 0.008 |
| ET/precipitation | 0.695 |

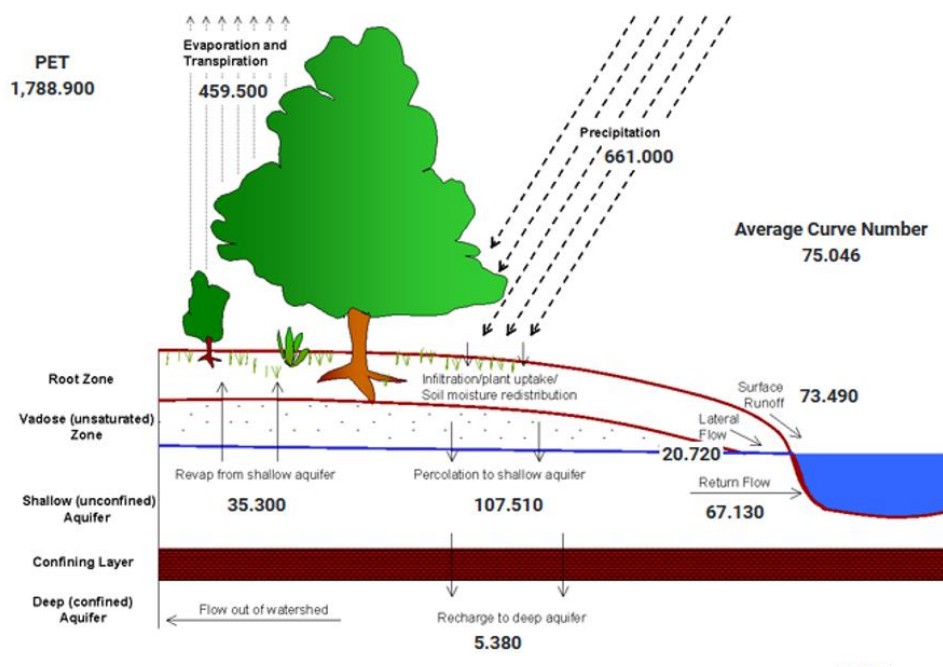

**Figure 3.** Average annual water balance of sub-basin 73 of the Pajeú-PE basin.

Climate change

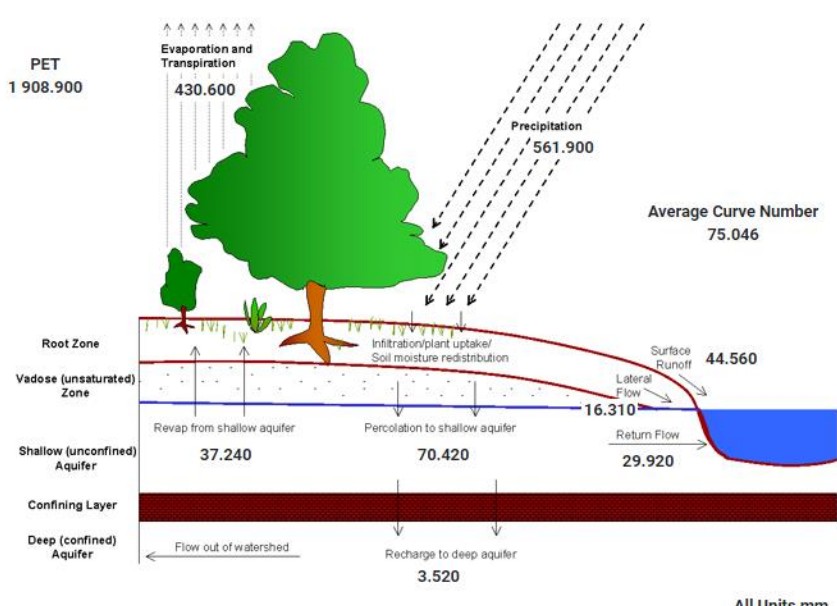

**Figure 4.** Average annual water balance, climate change situation.

*3.2. Actual*

The 25% of the year with flow above 1 m$^3$/s can be seen in Figure 5. The PISF increases in the Pernambuco state from 6 m$^3$/s to water security all year. But climate change decreases to 15% of the year, 36 days a year, which raises the question of whether the PISF project will only serve to mitigate climate change.

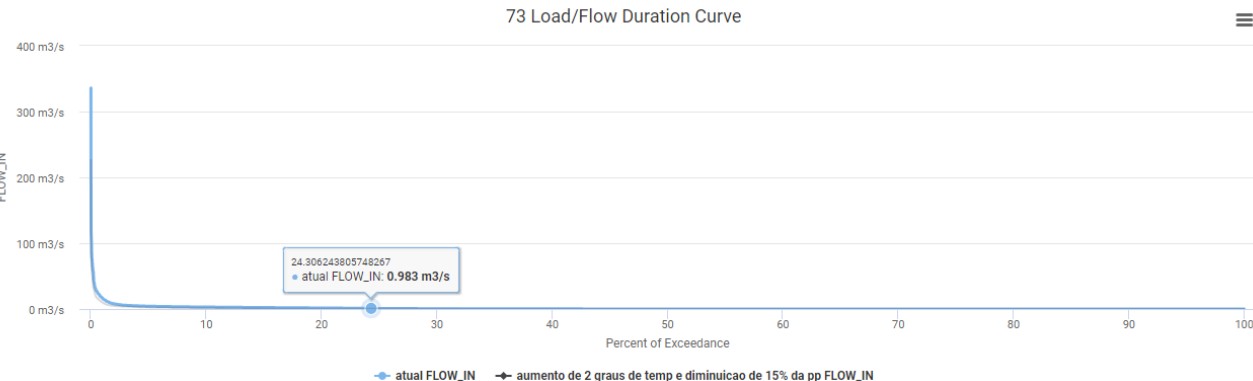

**Figure 5.** Flow duration curve in sub-basin 73, Pajeu basin.

Under the current average annual conditions, the vegetation of the sub-basin 73 under study undergoes seven days of average temperature stress, 70 days of water stress, 30 days of nitrogen stress and 43 days of phosphorus stress, as seen in Figure 6. Due to climate change, the days with temperature stress decrease to 2, the days with water stress increase to 83, the days with nitrogen stress drop to 18 and the days with phosphorus stress drop to 29, as displayed in Figure 7. That is, it is possible to perceive a high impact on the conditions of vegetation with climate change in the area. The decrease in the range of days with temperature stress can be fatal for different plant species in the area. The decrease in phosphorus and nitrogen can also impact the death of plant species, as well as the water requirement for family farming belonging to the area. These conditions can impair the quality of life for people living in the area. It is important to highlight that climate change causes different impacts according to land use, soil type and slope. For SPAS use, TC soil and a slope of 3–8 in this study, climate change can impact increasing biomass, as shown in Figure 8. This is more evident in areas with lower slopes, as seen in Figure 9. This may be due to these areas having greater water availability. The conditions of the soil, for example, soil RQ that represents the relative activities of aerobic and anaerobic microbial metabolism, and with a high slope that showed the importance of an increase in biomass, can be observed in Figures 10 and 11. The cropland/woodland mosaic, soil types which are TC—eutrophic soil and the slope of 0–3, in this land use, and the future decrease in biomass, can be seen in Figures 12 and 13 (3–8 slope).

Regarding the biomass in sub-basin 73 with conditions of CRWO-cropland/woodland mosaic, the soil types are RQ—Risk Quotient and there is a slope of 3–8, as seen in Figure 14, it is possible to see an increase in the future.

When the analysis averages and reaches its minimum and maximum, Figures 15–17 for CROW display an important increase in biomass.

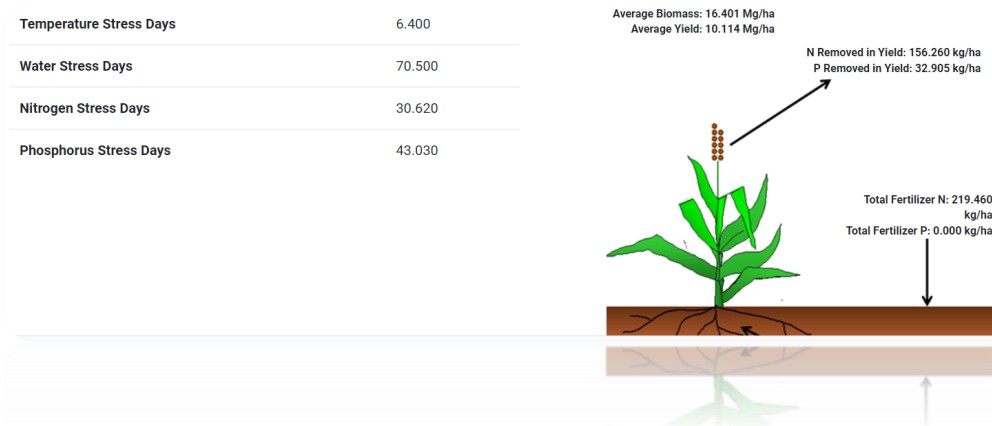

**Figure 6.** Annual environmental conditions of the vegetation, actual situation.

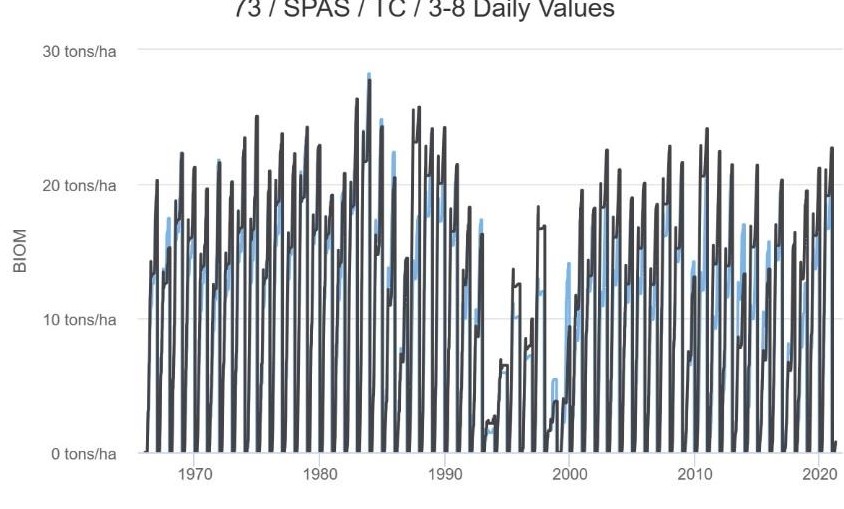

**Figure 7.** Annual environmental conditions of the vegetation, climate change situation.

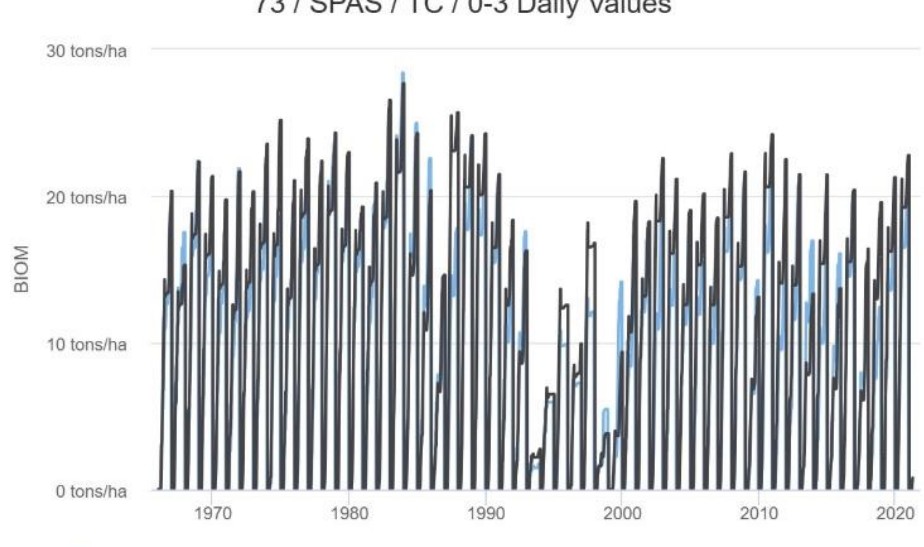

**Figure 8.** Temporal variation of biomass in sub-basin 73. SPAS-SPAS-summer pasture, soil types are TC—eutrophic soil and slope is 3–8.

73 / SPAS / TC / 0-3 Daily Values

**Figure 9.** Temporal variation of biomass in sub-basin 73. SPAS—SPAS-summer pasture, soil types are TC—eutrophic soil and slope is 0–3.

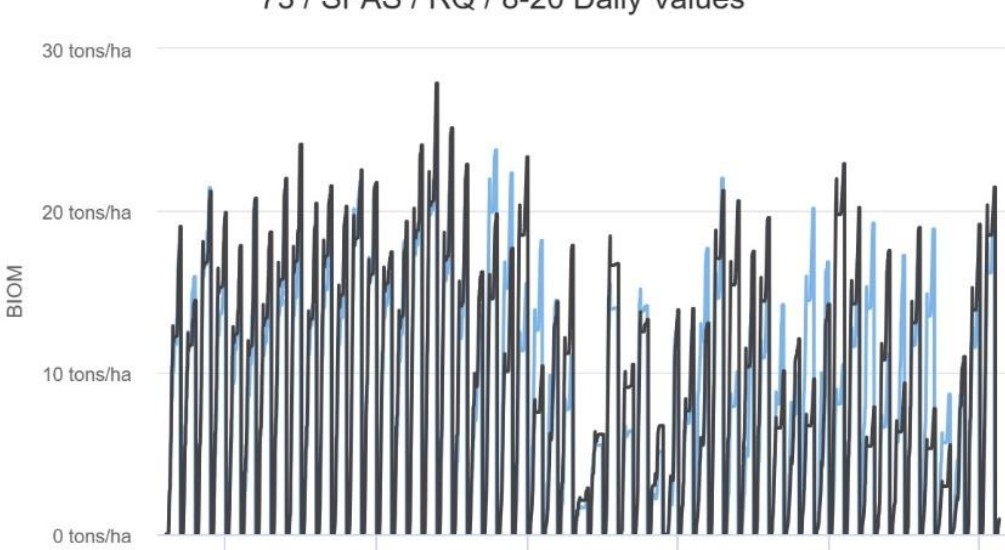

**Figure 10.** Temporal variation of biomass in sub-basin 73. SPAS—SPAS-summer pasture, soil types are RQ—Risk Quotient (represents relative activities of aerobic and anaerobic microbial metabolism) and slope is 8–20.

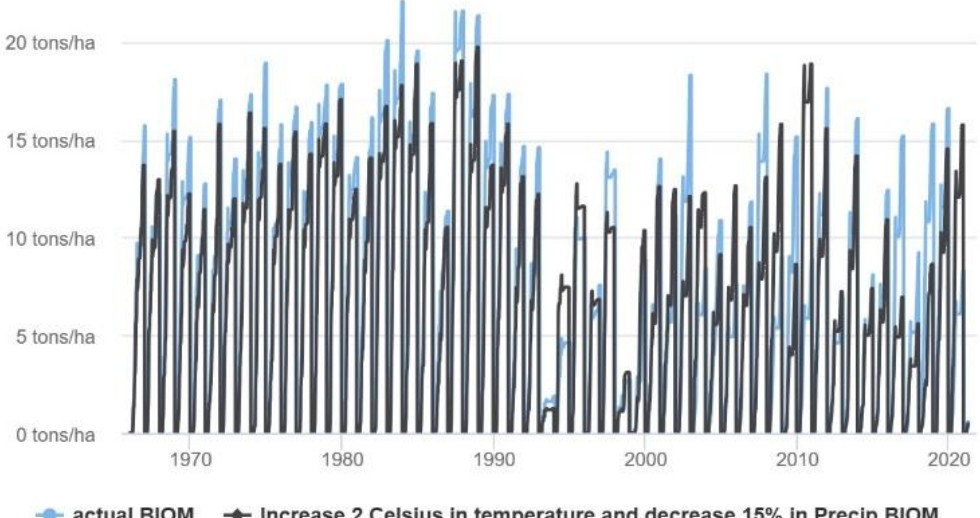

**Figure 11.** Temporal variation of biomass in sub-basin 73. SPAS—SPAS-summer pasture, soil types are RQ—Risk Quotient (represents relative activities of aerobic and anaerobic microbial metabolism) and slope is 20–45.

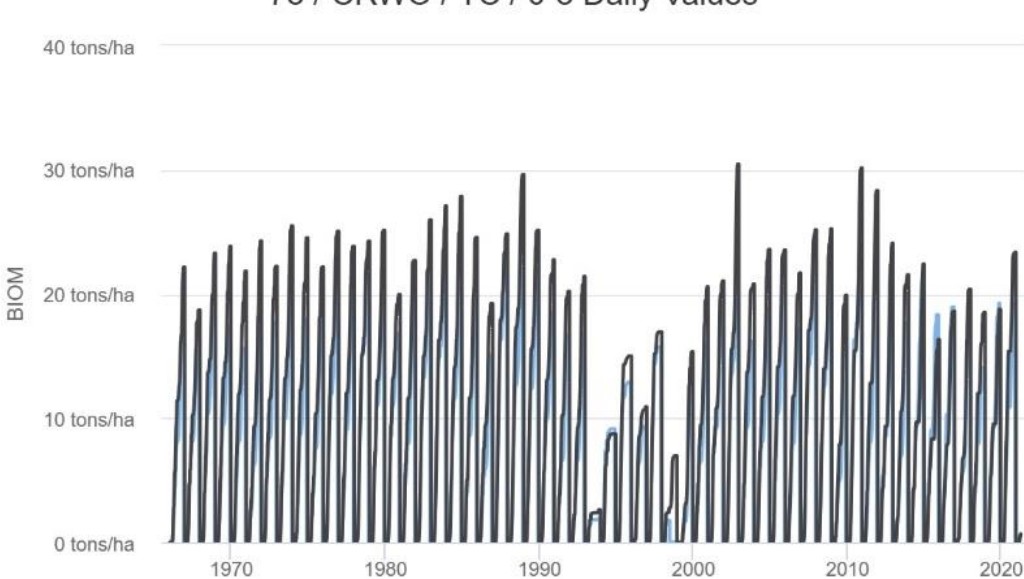

**Figure 12.** Temporal variation of biomass in sub-basin 73. CRWO—cropland/woodland mosaic, soil types are TC—eutrophic soil and slope is 0–3.

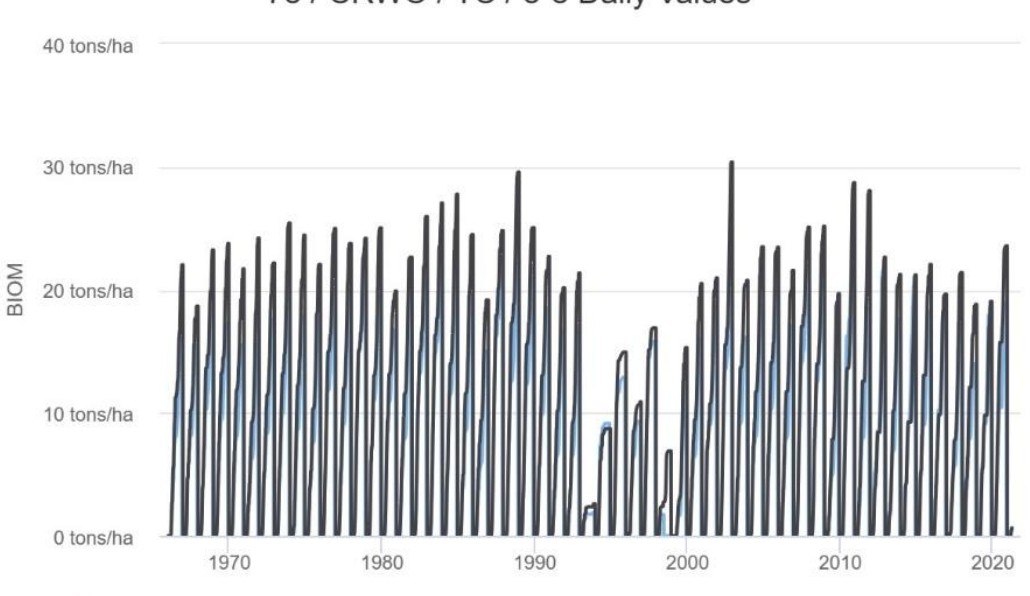

**Figure 13.** Temporal variation of biomass in sub-basin 73. CRWO—cropland/woodland mosaic, soil types are TC—eutrophic soil and slope is 3–8.

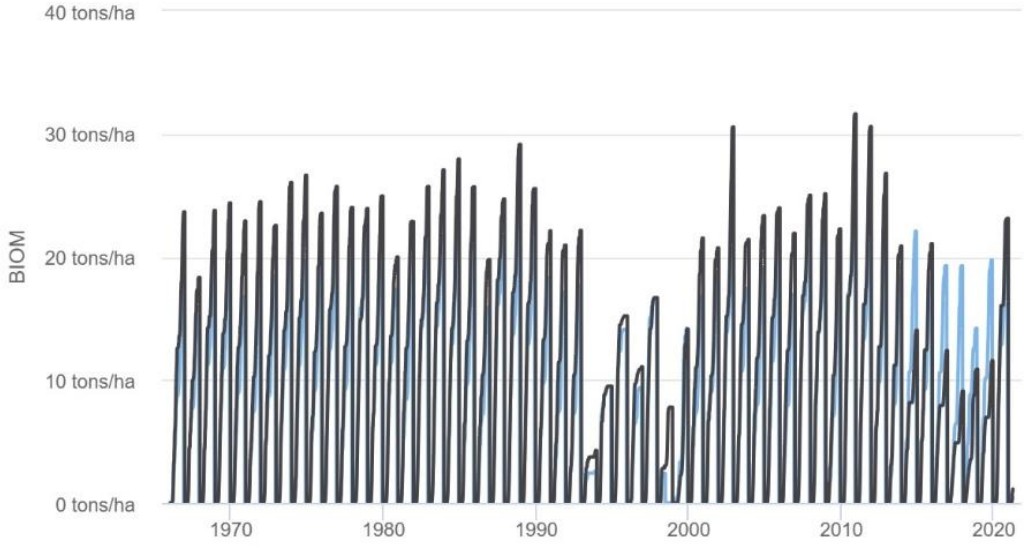

Figure 14. Temporal variation of biomass in sub-basin 73. CRWO—cropland/woodland mosaic, soil types are RQ—Risk Quotient (represents relative activities of aerobic and anaerobic microbial metabolism) and slope is 3–8.

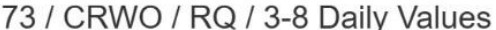

Figure 15. Daily averages biomass.

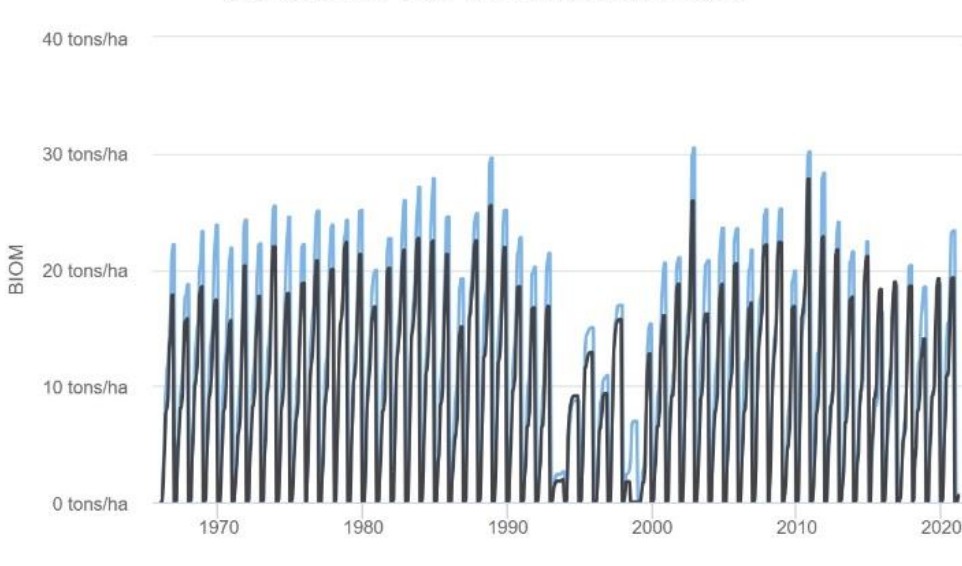

**Figure 16.** Daily maximum values biomass.

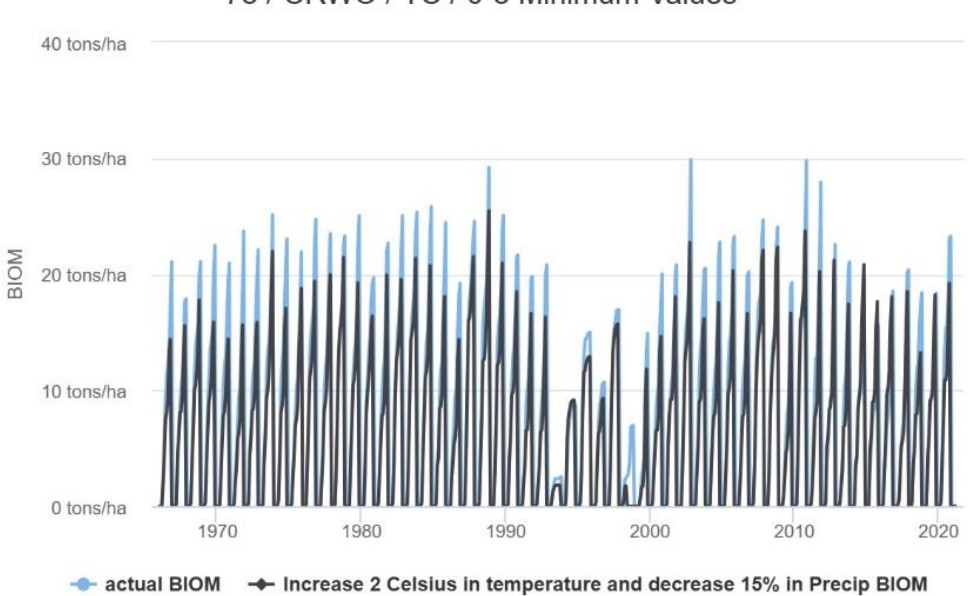

**Figure 17.** Daily minimum values biomass.

It is possible to observe the influence of the channel on the surface temperature. It is also possible to see the influence of green vegetation near the residences (Figure 18A–F). This green vegetation influences the surface temperature and consequently the temperatures of the residences, providing a better quality of life when related to temperature conditions. In view of what was presented in this study, namely that the areas of CROW with climate change will increase the biomass, it is believed that the surface temperature in the areas where there is an increase in biomass will be softened. The increase in water supply through the channel of the PISF-Integration Project of the São Francisco River will have an influence on the vegetation cover and an increase in biomass, especially in the CROW areas. Thus, it is concluded that climate change will have different impacts at small scales.

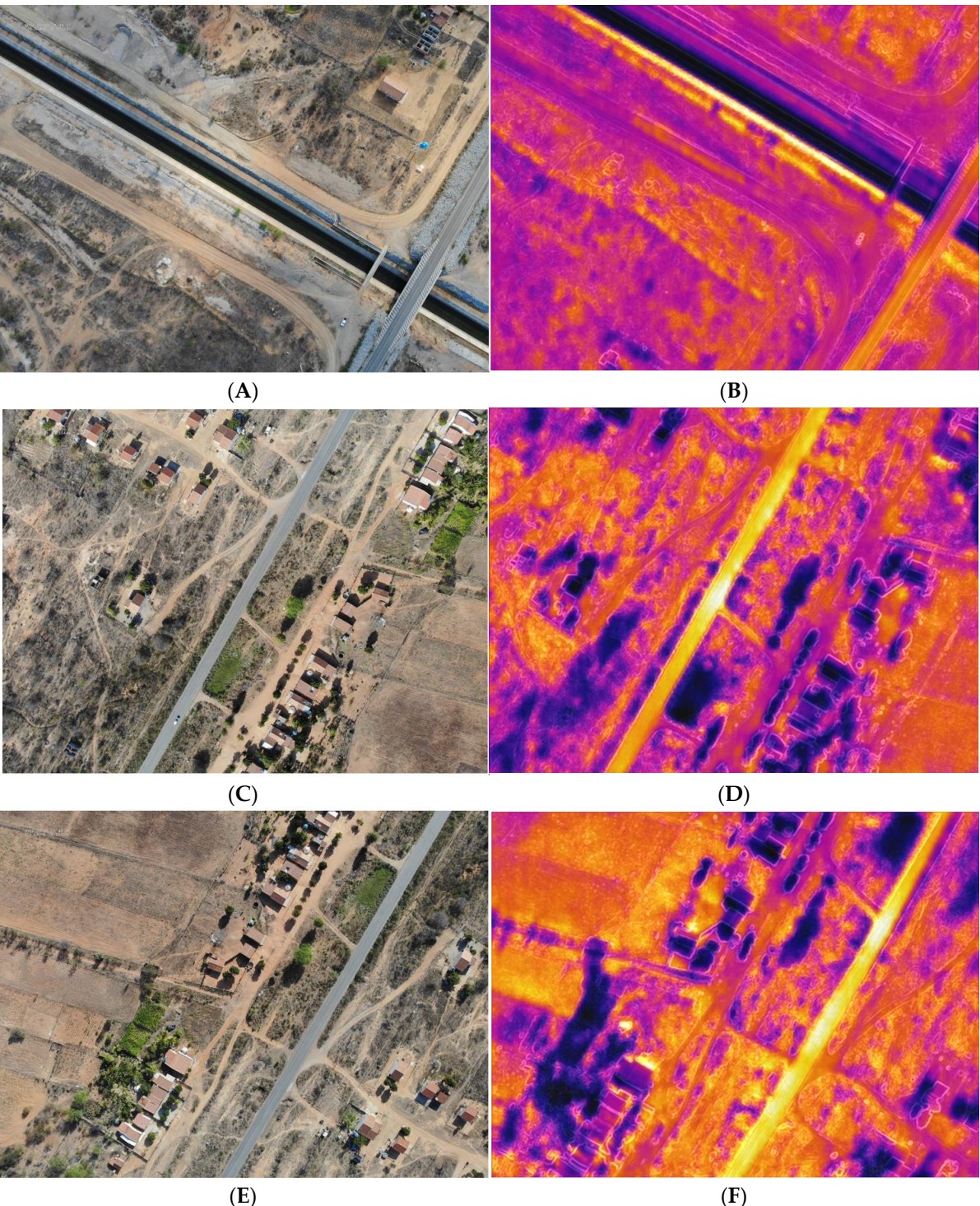

**Figure 18.** RGB and surface temperature images in small area of the sub-basin 73. (**A**) RGB. (**B**) Surface temperature. (**C**) RGB. (**D**) Surface temperature. (**E**) RGB. (**F**) Surface temperature.

## 4. Discussion

The findings of this study have revealed a deeply concerning state of water resources within [31] the investigated region, prompting the activation of an alert system, namely the SUPer system, which issues warnings and messages upon detecting a period of water stress exceeding 100 days. Prolonged episodes of water stress bear profound implications, particularly within a region characterized by severely constrained water availability. The protracted scarcity of water can precipitate a shortage of potable water, thereby adversely

affecting public water supplies and rendering the water unfit for consumption, culinary use and personal hygiene. Contingency measures, such as water rationing strategies involving usage restrictions and controlled water distribution, may be enforced in response to such circumstances.

Moreover, agriculture represents a cornerstone of livelihood for rural communities inhabiting the semi-arid region under examination. It serves as the primary income source for local families [32]. Extended periods of water stress pose a significant threat to agricultural endeavors, potentially resulting in crop damage, diminished agricultural production, escalated food prices and even surges in unemployment rates. Furthermore, the dearth of water resources may induce forced migration, a phenomenon that has already manifested in the northeastern region, as elucidated in the works of [33,34].

An intriguing outcome of this study is the projected increase in the average annual biomass. However, it is important to note that the efficacy of carbon sequestration will not remain consistent. Analogous findings have been corroborated in publications pertaining to forest ecosystems in the United States, as exemplified by [35]. The trajectory of forest recovery hinges upon two pivotal parameters: the asymptotic saturation of aboveground biomass and the age at which the stand attains half saturation. Both parameters are presumed to be contingent upon climatic conditions [36]. Across all forest types, elevated temperatures generally foster greater levels of aboveground saturated biomass while diminishing the age at which half saturation is achieved [37]. Conversely, augmented rainfall exerts a positive influence on both saturated biomass levels and the age at which half saturation is attained. The estimated parameters derived from the model provide quantifiable metrics for saturated biomass and the midpoint age, as elucidated by [35].

## 5. Conclusions

Therefore, it can be concluded that the decrease in plant cover in the area has a significant impact on the water system. This can lead to the loss of fertile soil, lower soil quality and more sediment in waterways as rainwater runs off. This increased runoff can carry soil particles, nutrients, pesticides and pollutants into water bodies, harming water quality.

It is also evident that water availability is decreasing, potentially causing a shortage of clean drinking water, which is essential for health and hygiene. Agriculture, crop production and livestock farming are also negatively affected, which can worsen food security and reduce income. This highlights the importance of monitoring water resources.

In summary, the study suggests that people's quality of life in this area is likely to decline due to reduced water availability and poor vegetation health. To address these issues, effective public policies are needed to mitigate the impact of climate change on water resources and the well-being of residents. Additionally, this study emphasizes the need for well-executed and closely monitored public policies to protect the environment and improve people's lives.

**Author Contributions:** Methodology, R.d.Q.M., J.D.G. and G.G.d.L.; Software, R.d.Q.M.; Validation, G.G.d.L. and J.D.G.; Formal analysis, G.G.d.L.; Investigation, G.G.d.L.; Resources, J.D.G.; Data curation, G.G.d.L. and R.d.Q.M.; Writing—review & editing, J.D.G. and G.G.d.L. All authors have read and agreed to the published version of the manuscript.

**Funding:** This research was funded by CNPq through the granting of financial resources to research project number 405853/2022-0 and by FACEPE through the granting of research project number APQ-0392-3.07/22.

**Acknowledgments:** The authors would like to thank the Universidade Federal de Pernambuco for the space given to carry out the research during the master's degree of the first author and CAPES.

**Conflicts of Interest:** The authors declare no conflict of interest.

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
