# Peer review of "Assessment of Water Availability and Environmental Influence on People’s Lives in a Small Basin in the Hinterland of Pernambuco, Using the SUPer and UAV"

_applsci, doi:10.3390/app132011255_

Round 1

Reviewer 1 Report

This MS focused on a small basin in Brazil and attempted to elucidate water scarcity-related issues. Global water resources with respect to the availability and resilience are of extremely high concern for water security and governance. However, the subject of this MS is really huge and the MS is not well organized. Dramatic modification is urgently suggested for authors including: i) to select an interesting scientific issue and propose it in a way the readers could easily understand; ii) to clearly state the data and methods which are needed to conduct this study; iii) to explicitly describe results from several aspects of the issue; iv) to initiate a full discussion on the results and provide reasonable interpretations; v) to draw convincing conclusions and offer advices for water management at reginal or watershed scale; and vi) to complete the manuscript in concise accurate authentic expressions.

 Some specific issues are listed as follows but not limited to:

 1) In the abstract, redundant sentences about the background are suggested to be deleted and the objective and main findings to be supplemented.

2) The keywords were not representative for the study theme.

iii) The unnecessary statements on background, local measurements and general knowledge are better to avoid in the Introduction, and literature review on the subject is crucial.

3) The objective is not quite consistent with the title and content.

4) Essential information on the study area such as geography, hydrology, geomorphology and environment ought to be provided in Materials and method and in Figure 1.

5) Is the study area Pajeu basin or a sub-basin of Pajeu basin?

6) The name of the basin, Pajeu basin or Pajeú basin, ought to be unique in the MS.

7) English is required for the contents in Lines 111 – 113.

8) The details on the data including collection, pretreatment and analysis need to be fully stated.

9) All approaches for model calibration and validation need to be explicitly elaborated.

10) The criteria for scenario setup ought to be clearly proposed.

11) How were the vegetation and the hydrology in the studied basin linked and how did they affect people’s lives?

12) The discussion and conclusions ought to be recast.

The authors are suggested to carefully check the expressions of the MS to avoid grammatical errors and spelling mistakes. Besides, native speakers should be invited to polish the language of the MS.

Author Response

All suggestions were met.

Reviewer 2 Report

Excellent manuscript. Accepted in current form

Author Response

Thank you.

Reviewer 3 Report

This manuscript provides a model framework for assessing water availability and environmental impacts on the lives of residents in small watersheds, an interesting topic with practical significance.     But there are some problems that need to be considered.     The details are as follows:

(1) Abstract.     The background is too long, taking up about half of the abstract, which is inappropriate, and the authors are advised to further simplify.

(2) Introduction.     There are not enough explanation to solve the existing problems in the current research.

(3) Materials and Methods.     The logic is not clear, so it is recommended to show it in sections.

(4) Figure.     From Figure 8 to Figure 17, the color contrast used is not high, and the perspective effect is poor

(5) Discussion.     Discussion should be a very important part, but the discussion in this document seems too simplistic.

Minor editing of English language required

Author Response

All suggestions were met.

Reviewer 4 Report

Please revise the English grammar.

Please review, edit and revise the English grammar.

Author Response

Attached the paper with corretions.

Reviewer 5 Report

There is little to no description of the model that you use. Do you expect the readers to know what SWAT2012 is or the SUPer model? These need to be described and explained much better than what was done in the paper. 

Sections 4 and 5, Discussion and Conclusions might as well not exist. Both of these sections are not even close to good enough for the paper to be published. There is a lot of information and findings in your paper that you can cover in these sections. It's as if you got to these sections and said, we are tired, that's enough. Neither Section is appropriate. 

Fine, minor editing

Author Response

comments:

There is little to no description of the model that you use. Do you expect the readers to know what SWAT2012 is or the SUPer model? These need to be described and explained much better than what was done in the paper. 

Sections 4 and 5, Discussion and Conclusions might as well not exist. Both of these sections are not even close to good enough for the paper to be published. There is a lot of information and findings in your paper that you can cover in these sections. It's as if you got to these sections and said, we are tired, that's enough. Neither Section is appropriate. 

answer: The changes were made. The model was well described and the
discussion and results improved

Round 2

Reviewer 1 Report

Compared with the original submission, authors have made modifications in sections of abstract, introduction, materials and method and discussion in this revised manuscript. However, further efforts are strongly required to improve it. Authors are suggested to put greater emphasis on the logicality of the study. The manuscript would not be reconsidered for publication until the proposed issues are completely addressed. The specific comments are listed as follows.

(1) Too much background and significance were declared, while little was on main findings. How water variability affecting people's lives ought to be clearly stated.

(2) Very common information were presented in the section of introduction. But what made you to choose such a scientific theme was not interpreted, and this is the critical point of your study. Besides, literature review ought to be improved and more summaries on previous studies related to water availability affecting people's lives should be supplemented.

(3) In the section of materials and method, the SWAT and statistical analysis were stated. But it was little hard to understand what these techniques were used for.  What is 'people's lives' and how could it be qualitatively and quantitatively described. The point of your study rests on a match between water availablity and people's lives. It's critical to quantify these two parameters. Furthermore, it would be better to expound these methods in a problem-orientated way.

(4) I wonder how values of some parameters were obtained in this study such as strees days, RQ, historical biomass, simulated biomass and so on.

(5) Extremely brief discussion was presented which is very inappropriate.

Authors are strongly recommended to ask a native speaker to better the quality of English language. Syntax errors and non-English expressions ought to be avoided.

Author Response

Please see the attachment."

Reviewer 3 Report

The author has made appropriate revisions and responses. I think appropriate formatting adjustments and graphics resolution optimization are required.

Author Response

In the new version, the requested adjustments have been made. Thank you for the contributions

Reviewer 4 Report

Please see attached comments.

Paper still needs extensive English grammar proofing and editing.

Author Response

In the new version, the requested adjustments have been made. Thank you for the contributionsIn the new version, the requested adjustments have been made. Thank you for the contributions

Reviewer 5 Report

The paper has been sufficiently upgraded in my opinion

Author Response

 Thank you for the contributions